# Roles of p53 Family Structure and Function in Non-Canonical Response Element Binding and Activation

**DOI:** 10.3390/ijms20153681

**Published:** 2019-07-27

**Authors:** Bi-He Cai, Chung-Faye Chao, Hsiang-Chi Huang, Hsueh-Yi Lee, Reiji Kannagi, Jang-Yi Chen

**Affiliations:** 1Department of Biology and Anatomy, National Defense Medical Center, Taipei 11490, Taiwan; 2Institute of Biomedical Sciences, Academia Sinica, Taipei 11529, Taiwan

**Keywords:** p53, p63, p73, noncanonical sequence, dimer, tetramer

## Abstract

The p53 canonical consensus sequence is a 10-bp repeat of PuPuPuC(A/T)(A/T)GPyPyPy, separated by a spacer with up to 13 bases. C(A/T)(A/T)G is the core sequence and purine (Pu) and pyrimidine (Py) bases comprise the flanking sequence. However, in the p53 noncanonical sequences, there are many variations, such as length of consensus sequence, variance of core sequence or flanking sequence, and variance in number of bases making up the spacer or AT gap composition. In comparison to p53, the p53 family members p63 and p73 have been found to have more tolerance to bind and activate several of these noncanonical sequences. The p53 protein forms monomers, dimers, and tetramers, and its nonspecific binding domain is well-defined; however, those for p63 or p73 are still not fully understood. Study of p63 and p73 structure to determine the monomers, dimers or tetramers to bind and regulate noncanonical sequence is a new challenge which is crucial to obtaining a complete picture of structure and function in order to understand how p63 and p73 regulate genes differently from p53. In this review, we will summarize the rules of p53 family non-canonical sequences, especially focusing on the structure of p53 family members in the regulation of specific target genes. In addition, we will compare different software programs for prediction of p53 family responsive elements containing parameters with canonical or non-canonical sequences.

p53 is a tumor suppressor gene that can be directly active at least 350 genes which are involved in cell apoptosis, cell cycle arrest, autophagy, metabolism, DNA repair, translational control, and feedback mechanisms [1] (a list of genes can be found in IARC TP53 Database: http://p53.iarc.fr/TargetGenes.aspx). p63 and p73 are both p53 family members. In p53, p63, and p73 single knockout mice, p53KO mice appear tumor prone but p63KO or p73KO mice do not [2,3,4]. In p53^+/−^;p63^+/−^ or p53^+/−^;p73^+/−^ double heterozygous mice develop a higher frequency of metastatic tumors than p63^+/−^;p73^+/−^ mice [5] (the relevant mouse models are summarized at: http://p53.free.fr/p53_info/Mouse_model/p53_mouse_models.html). Therefore, each member of the p53 family can activate similar genes and also have specific target genes. Besides the canonical sequences of p53 family response elements, noncanonical sequences have also been identified. It has been found that p63 and p73 have more tolerance to bind and active these noncanonical sequences. In this review, we will summarize the rules of p53 family noncanonical sequences, focusing on the structure of p53 family members in the regulation of specific target genes. In addition, we will compare different software programs for prediction of p53 family responsive elements containing parameters with canonical or noncanonical sequences.

## 1. Noncanonical Sequences of the p53 Family Regulation Sites

### 1.1. Different Lengths of Consensus Sequence

The p53 canonical consensus sequence is a 10-bp repeat of PuPuPuC(A/T)(A/T)GPyPyPy (Figure 1A) [6]. One classical experiment evaluated the different lengths of consensus sequence that could also be bound by p53 [7]. Incubation of the p53 protein with labeled p53 whole-site and different lengths of consensus sequence unlabeled competitors were analyzed by EMSA. The competitors of the whole-site, half-site, but not the quarter site could compete with the whole-site binding [7]. In addition, 1.5× half-site competitor also could compete with the whole-site binding [8]. Therefore, the p53 half-site, 1.5× half-site and whole-site are all able to be bound by p53. The reporter constructs of these sequences show that 1.5× half-site and whole-site could be activated by p53, while p53 half-site is not activated [8]. The 1.5× half-site contains two types of sequences: consecutive 1.5× half-site sequences, such as GAA**CAAG**TCCGGG**CA**TaTgT on *PCNA* promoter with the first, second and third quarters [9]; and nonconsecutive 1.5× half-site sequences, such as AGG**CT**TcTTgTTCAGG**CTTG**CTC on *JAG1* promoter, which contains the first, third and fourth quarters [10]. The consecutive 1.5× half-site is activated by p53 more strongly than the nonconsecutive 1.5× half-site [8]. p53retriever (http://tomateba.github.io/p53retriever/) is a R programming language that searches the p53 putative response elements with noncanonical half-sites and 1.5× half-sites and also ranks their predicted transactivation potentials into five different grades of functional scores from five (highly functional) to one (unlikely functional) of each response element [11]. A C-to-T single-nucleotide polymorphism (SNP) in the vascular endothelial growth factor receptor 1 (also called *Flt1*) appeared as a p53 half-site GGACA**t**GCTC [12]. The *Flt1*-T SNP promoter can be activated by p53 because the up-strand contains an estrogen responsive element (ERE), and p53 almost no longer activates the *Flt1*-T SNP promoter with mutation of ERE [13]. In addition, promoters containing both the half-site of ERE and the half-site of p53 could greatly expand the p53 regulation network [9,14]. High expression levels of p53 can activate a half-site only that contains core sequence CATG but not CTAG [15]. Conversely, p63 was able to activate a p53 half-site on *KRT14* promoter, and p53 could activate the mutants with extended sequence as whole-site of *KRT14* promoter [16]. Because p53 could repress the *KRT14* promoter through the SP1 binding site [17], the mutants with extended sequence, such as 1.5× half-site on *KRT14* promoter, were still not activated by p53 [16].

### 1.2. Variance of Core Sequence

The *EVPL* promoter contains two whole-sites of p53 consensus sequence [18]. Site 1 could be bound by both p53 and p63γ, but site 2 was only bound by p63γ but not by p53. Site 2 GTGCA**G**GAGGAGGCATGAGT contains one CA**G**G core sequence. The *SMARCD3* promoter contains one p53 consensus sequence GGGC**G**TGCAGATGCAAGCAC which contains one C**G**TG, and this is only activated by p63γ but not by p53 [18]. The mutant GGGC**a**TGCAGATGCAAGCAC was activated by both p53 and p63γ. In addition, *PKC-δ* promoter contains three p53-like responsive elements, and the first one GGGGAGTCCCGGGC**G**TGGGT is a 1.5X half-site. To mutate the first p53-like responsive element sequence only reduced the p63 but not p53 mediated transactivation function of the *PKC-δ* promoter [19]. Therefore, p63 prefers the C**G**TG core sequence (Figure 1B) [18]. On the other hand, the *GDF15* promoter contains two whole-sites of p53 consensus sequences. Site 1 could be bound by all p53 family members, but site 2 could only be bound by p53 but not p63 or p73 [20]. When the site 2 AGCCATGCCCGGGCA**A**GAAC was mutated to AGCCATGCCCGGGCA**t**GAAC, it could be bound and activated by TAp73β [20]. In addition, AGCCATGCCCGGGCA**t**G**cc**C could be bound and activated by both TAp63β and TAp73β [20]. Therefore, in addition to variation in core sequence, the variation in flanking sequence is also a key point in determining p53 family specific targeting.

### 1.3. Variation in Flanking Sequence

According to EMSA binding experiments, all core sequences CA**G**G and C**G**TG and their palindrome sequences C**C**TG and CA**C**G can be bound by p63 [18]. TAACATGTTT or TAACTTGTAT or TAAC**C**TGTAT or TAAC**G**TGTAT bound to p63 much more strongly than AGGCATGATG or GAGCTTGATT or GACC**C**TGACC or CCCC**G**TGCAC in EMSA experiments [21]. Therefore, p63 prefers AT-rich flanking sequences rather than general GC rich in purine (Pu) and pyrimidine (Py) bases that comprise the flanking sequence for p53 or p73 (Figure 1B) [21]. p73β binds to *p21* and *PUMA* promoters more strongly than p53. Analyses of the flanking sequences of *p21*- and *PUMA*-binding sites indicated that five conserved bases PuPuPuC(A/T)(A/T)G**TCC**PuPu**A**C(A/T)(A/T)G**T**PyPy have tighter binding affinity when bound by TAp73β than by p53 [22], but there is no difference between PuPuPuC(A/T)(A/T)G**CTT**PuPu**G**C(A/T)(A/T)G**C**PyPy or PuPuPuC(A/T)(A/T)G**AGG**PuPu**T**C(A/T)(A/T)G**A**PyPy bound with TAp73β or p53.

### 1.4. Base Number of AT Gap

The *14-3-3σ* promoter has a p53 regulation site which contains a responsive element GtAGCAttAGCCCAGACATGTCC. The first core sequence is CAttAG and the second is CATG. The mutant GtAGCAttAGCCCAGACATcTCC can be activated by p53 as a 1.5-fold half-site as GtAGgAttAGCCCAGACATGTCC mutant [23]. Therefore, CATTAG is a functional core sequence that contains four AT gaps. In addition, a six-base AT gap CATATATG core sequence was the maximum AT gap in the p53-responsive element that could be upregulated by p53 and p63 (Figure 1B) [24]. AT gaps of 3, 5, 7, or 8 bases were not bound and activated by p53 (Figure 1C) [23,24].

## 2. Other Sequence Variation Also Influences p53 Family Activity

### 2.1. Base Number of Spacers

The crystal structure of p63 DNA-binding domain tetramer can bind to 1-bp overlapping response element, AAACATG**TTAAA**CATGTTT [25]. This means that only five bases between two core sequences could be activated by p63. In reporter assays, both p53 and TAp63γ could activate six bases between two core sequences, but only TAp63γ could activate five bases between two core sequences [24]. Therefore, p63 could bind and activate a consensus sequence with a -1 spacer. On the other hand, p53 and p73 may tolerate the space number on a case-by-case basis. Two half-sites contain a one base spacer reducing 90% of p73β transactivation activity, and insertions of 2 or 4 bp spacers reduce the transactivation response to background levels [26]. p53 could tolerate 1, 2, and 4-bp spacers without an obvious drop in transactivation activity [26]. On the other hand, *IL-4Rα* intron 2 contains a 104 bp responsive element with a half-site 5 bp spacer, half-site 9 bp spacer, whole-site–half-site 9 bp spacer, and half-site 1 bp spacer whole-site, which could be activated by p53 and p73β. Deletion of all the spacers as in the resulting 80 bp responsive element dramatically reduced the p73β mediated transactivation but had no influence on p53 activation [27]. Therefore, the spacer sequence on the *IL-4Rα* responsive element is important for p73 activation. Two half-sites with a 10 bp spacer in-between could be activated by p53 at ~80% activity compared to the no spacer whole-site, but 4, 5, 13, 14, and 15 bp spacer only left ~10% activity [8,28]. The DNA-binding domains of p53 represent loop–sheet–helix motif regions that make specific base contacts in the major grooves [29], and 10 bp spacer will make the monomer A and C face the same direction as the major groove and monomer B and D face the another direction of the groove opposite to A and C as indicated in 0-spacer response element in Figure 2A [30,31]. A similar study used a 20 bp spacer and also found the tetramer Kd to be as low as that of 10 bp [32]. Binding of p53CT (aa 94–360) to a 20 bp spacer was found to have two patterns of tetramer binding as fully specific and hemispecific types. A fully specific tetramer binding pattern means two p53 dimers bind to two half-sites (Figure 2B). A hemispecific tetramer binding pattern means that only one of its p53 dimer binds to a half-site, but the other dimer binds to the nonspecific spacer DNA [32].

### 2.2. Cruciform DNA

Besides the serial rules influencing binding and function of the p53 family, the two half-site sequences with perfect inverted repeats or partially inverted repeats can form cruciform DNA. For both p53 and p73 it has been proved that the cruciform consensus sequence has much higher transactivation activity than the noncruciform consensus sequence [33,34,35].

## 3. Structure of p53 Family Members and Its Relationship to Their Target Sequences

### 3.1. Structure of p53

p53 contains an N-terminal transactivation domain (aa 1–50 in murine; aa 1–54 in humans), middle DNA binding domain (aa 80 to 320 in murine; aa 83 to 323 in humans), a C-terminal oligomerization domain (aa 315 to 350 in murine; aa 323 to 355 in humans) and a nonspecific DNA binding domain (aa 360–390 in murine; aa 363–393 in humans) [36,37]. Each p53 monomer binds to the one-quarter-site by the middle DNA binding domain [8]. Following the DNA binding domain is a turn and an β-sheet which dimerize the two monomers to from the dimer [38]. The following β-sheet by a turn and an α-helix forms the dimer–dimer interface to compose the tetramer [39]. The murine p53 aa 1–320 segment as a monomer could start to have a specific binding to a half-site, 1.5× half-site and whole-site [8]. p53 binds to the canonical consensus sequence of a p53 whole-site in tetrameric form. Tetrameric p53 also can bind to the p53 half-site by one of its dimers. The single amino acid mutant of human p53 L344A or L348A disrupts the dimer–dimer interface to form the dimer form only [40,41].

### 3.2. Structure of p63 and p73

p63 and p73 also have a N-terminal transactivation domain, a middle DNA binding domain, and C-terminal oligomerization domain [42]. Some isoforms of p63 and p73 produced by RNA alternative splicing have an additional SAM domain which follows the oligomerization domain and is involved in protein–protein interactions [43]. The entire SAM domains of TAp63α and TAp73α contain five helices, and TAp63β and TAp73β contain a partial SAM domain [44,45]. Because MDM2 is an E3 ligase of p53, it does not ubiquitize and degrade p63 and p73 [46,47,48]. The SAM domains are required for degradation of α and β isoforms of p63 and p73 by Fbw7 and FBXO45 E3 ligases respectively [49,50]. TAp63α and TAp73α contain an inhibitory domain, which follows the SAM domain [51], and the inhibitory domain can interact with the transactivation domain to block the transactivation function of TAp63α and TAp73α [52,53,54].

p63 can form a dimer only in oocytes which contain high expression of TAp63α [55]. The TAp63 isoform, which is constitutively expressed in oocytes, is essential in the process of DNA damage-induced oocyte death not involving p53 [56]. p53 and p73 generally exhibit low expression in normal somatic cells and they are only induced by DNA damage agents or stress signals [57,58,59,60]. TAp63α is highly expressed as a closed dimer form, which has low activity because the C-terminal inhibitory domain and N-terminal transactivation domain both interact with, and block, the oligomerization domain [61]. When oocytes sense DNA damage, TAp63α is phosphorylated by kinases such as CHK2 and CK1 to trigger a change in p63 conformation from a closed dimer to open dimer to adopt an active tetramer form [62,63]. This phosphorylation-mediated tetramerization of TAp63α cannot revert to dimer form by λ-Phosphatase [55]. Oligomerization domain of p63 contains an additional helix as the second helix which is not present in p53, and this helix locks TAp63α in its tetrameric form after phosphorylation and no longer comes back to dimer form [55]. A TAp63αFTL, which contains the mutation of F605A, T606A, and L607A, acts as an opened dimer to form a tetramer, but it becomes a dimer only after deletion of the additional helix [55]. p73 also has an inhibitory domain and a second helix within the oligomerization domain, and deletion of this helix could cause p73 to become the dimer only [64]. However, TAp73α does not form a closed dimer, and it keeps a constitutively open conformation to form a tetramer as well as p53 [60,65,66].

### 3.3. The Difference in the DNA Binding Domains of p53 Family Members

DNA-binding domains of p53 are loop 1 (K120)–sheet 10 (C277)–helix 2 (R280) motif regions that make specific base contacts in the major grooves [29], and the representing motifs on L1-S10-H2 are K138, C297, and R300 in p73 and K149, C308, and R311 in p63 [42]. R280 of p53 and R311 of p63 form hydrogen bonds with position 7 as a G on a half-site [67,68]. In addition, C277 of p53 and C308 of p63 form van der Waals interactions with position 8 as a T [68]. K120 of p53 forms hydrogen bonds with the position 2 as a G [67,69], but the structure shows that the bases at positions 1, 2, 9, and 10 do not interact with p63 [68]. A SELEX study showed that p53 and p63 prefer a T in position 8 (p53: 64%; p63: 87%), and only p53 but not p63 prefers a G in position 2 (p53: 73%; p63: 45%) [70]. This is consistent with the structure of protein-DNA interaction. p53 also interacts with DNA minor groove through R248. This residue is needed to stabilize the Hoogsteen base pairs to narrow the minor groove [71], and similar narrowing also occurs through R279 on p63 [68]. *MDR1* is a p63 and p73-specific activation gene which cannot be bound and activated by p53, but exchange of the p53 DNA binding domain with the p63 or p73 DNA binding domain could active *MDR1* promoter [72]. Therefore, the key residues within the DNA binding domain that are different in each p53 family member should make a contribution to their specific binding, although the DNA binding domain is the most conserved domain compared to the transactivation or oligomerization domain [42]. An interdimer double salt bridge can be formed by R180 and E181 on the p53 DNA binding domain [73], and the key residues are R209 and L210 present on p63 which cannot form the salt bridge [74]. Mutation of the p63 DNA binding domain from L210 to E210 can form the interdimer double salt bridge which can dramatically enhance the binding affinity to the binding sites on *p21* and *BAX* promoters with a similar Kd to p53 [74].

### 3.4. Nonspecific Binding Domain in p53 and p73

p53 with deletion of the nonspecific domain binds to 4 AT gap and 6 AT gap, but the binding is much weaker than towards the 2AT gap [23,24]. It is necessary to avoid nonspecific binding in each p53 family member in experiments. Indeed, the wild type of p63 for DAPA or wild type p73 for EMSA still have nonspecific binding [10,69]. The nonspecific binding activity of p53 is acquired by the many positive charge amino acids, arginine or lysine, in its C-terminal domain [75,76]. The isoelectric points (pI values) of the last 30 aa of p53 is 10. Several isoforms of p73 have also been studied [77]. The last 30 aa of TAp73γ is composed of 8.77 pI, and the last 76 aa of TAp73γ is composed of 11.76 pI. The last 30 aa of TAp73α, β and δ have isoelectric points between 4.4–7.01. Both p53 and TAp73γ bind to short oligonucleotides (28 bp) with the p53 consensus sequence while showing no binding in EMSA because of a sliding off effect to drop down on DNA. p53 and TAp73γ bind to long oligonucleotides (66 bp) having a p53 consensus sequence with weak specific binding and huge nonspecific binding [77]. On the other hand, TAp73α could bind to both short and long oligonucleotides without nonspecific binding. In addition, deletion of the last 30 aa of p53 as p53Δ30 was able to remove both the sliding-off effect and nonspecific binding, and TAp73α + 30, which is the TAp73α added the last 30 aa from p53, could cause the sliding-off effect and nonspecific binding [77]. p53Δ30 binding affinity for TGGCATGTCCCGACTTGTTA is similar to TAp73β and larger than p53 [22]. Although the C-terminal nonspecific binding domain has a negative effect to block the p53 specific binding to linear DNA, it could enhance the p53 binding to microcircle DNA or cooperate with HMGB1 in binding to bent DNA [78]. c-Abl could interact with both the oligomerization domain and C-terminal nonspecific binding domain, which enhances p53 binding activity, while c-Abl is no longer able to enhance p53 binding activity after deletion of the C-terminal nonspecific binding domain [79]. In addition, the posttranslational modifications or mimic mutant of the C-terminal nonspecific binding domain amino acids could alter the binding affinity to DNA [80,81,82]. One p53 phosphorylation mimic S392E was able to enhance target recognition probability [83], and the nonspecific binding domain was able to perceive the salt concentration to restrict the association of two primary dimers into a tetramer in low salt concentration or to restrict hopping of DNA binding domain in high salt concentration [84,85,86]. At low salt concentration (50 mM ion concentrations), p53 tetramer of all four DNA binding domains and nonspecific binding domains slides and scans long DNA, but at high salt concentration (200 mM ion concentrations), three nonspecific binding domains were bound to DNA, while only one or two DNA binding domains remained bound to DNA [85]. p53 might completely dissociate from DNA without nonspecific binding domain in the scanning processing on long DNA at very high salt concentrations. Therefore, the C-terminal nonspecific binding domain can have a negative or positive role in p53 specific binding.

### 3.5. The p53 Family Structure and Noncanonical Sequence Relationship

A dimeric form of p53 mutant p53L344A can also bind to the p53 half-site [7]. In addition, the tetrameric p53 binding to a consecutive 1.5× half-site as first, second, and third quarter-sites is much better than nonconsecutive 1.5x half-site as first, second, and fourth quarter-sites [8]. Tetrameric p53 (p53Δ363-393) could bind to a 1.5x half-site and a p53 whole-site with a 4AT gaps sequence, but could not bind to a p53 half-site with a 4AT gap [23]. In addition, the dimeric form of p53 mutant p53L344AΔ363–393 could not bind to a p53 half-site with a 4AT gaps sequence [23]. Therefore, the monomer C on the tetramer associated with the third quarter is absolutely needed to help the tetramer in binding to the core sequence with 4AT gaps (Figure 2C). In p63 and p73, no further experimental information is available regarding whether any single amino acid substitution can mediate formation of the pure dimer, which is formed by p53L344A in the case of p53. To determine this, the following experiments will be needed. First, it will be necessary to define the nonspecific binding domain of p63 and p73 isoforms. Then it will be necessary to delete both the nonspecific domain and the second helix in the p63 and p73 oligomerization domain, which will generate the pure dimer form of p63 and p73 with specific binding activity. When these dimers are subjected to the binding study against noncanonical sequence, it will fully clarify the activity of p63 or p73 dimer and their binding pattern to the noncanonical sequence. This may offer more information and make clearer why p63 and p73 are more tolerant than p53 to bind and activate these noncanonical sequences.

## 4. p53 Family-Specific Genes and Prediction

Prediction software programs for p53 family responsive element sequences contain the motifs of p53 only or of each respective p53 family member [87] (Table 1). Several also have the half-site only as a parameter for prediction. When we obtain all the information about the structure and sequence tolerance relationships for all the p53 family members, maybe we will be able to understand why each p53 family member regulates its own specific genes (Table 2). For example, new findings about glycan gene targets *NEU4* and *FUCA1* show that p73 can upregulate *NEU4* promoter, but p53 cannot [88]. On the other hand, p53 can upregulate *FUCA1* promoter [89], but p73 cannot [90]. In addition, *p57Kip2* could be upregulated by p63 and p73 but not by p53 [91,92], but the p63 and p73-specific binding site on *p57Kip2* has still not been identified [93,94]. More details about the noncanonical rules for the binding site of each p53 family member and more accurate software for prediction will make it easier to identify the specific regulation sites of p53, p63, and p73. In addition, according to the papers published between 1992 to 2016, 71% (246/346) of individual p53 target genes published were upregulated, 26% (91/346) were downregulated, and 3% (9/346) were up- or downregulated [1]. Analyses of p53-responsive element motifs from 123 validated activating p53 response elements and 39 validated repressing response elements found that the sequences were quite similar, RRRCWWGYYYRRRCWWGYYY for activation and RRXCXXGXYXXRXCXXGXYY for repression (R is A or G, W is A or T, Y is C or T, X is A or G or C or T) [95]. Therefore, some of the p53 prediction software based on SELEX or ChIP-seq data without further comparison to RNA seq with up- or down regulation information may have bias and include the possibility to predict targets that were not only activated but also repressed by p53.

## 5. Concluding Remarks

p53 was identified in 1979 has been known for 40 years [96,97]. p63 and p73 were identified in 1998 and 1997, respectively, and have been known for over 20 years [98,99]. However, to date, our knowledge of the roles of the p53 family members besides their role in cancer control is limited [60,100,101]. Because the binding site of the canonical sequence is very similar for each member, identification of the details of rules of noncanonical sequences and knowledge of the structural differences between each member may further provide more accurate parameters for developing software for the prediction of specific target sites for each family member. For example, the core sequence variance, which is frequency been identified for p63- or p73-specific genes (Table 2), should be distinguished by prediction software. Several studies have mentioned that p53 with no further post-translational modification could only activate some genes (like *MDM2* and *PIRH2*) [102,103]. p53 becomes a partially activated form through phosphorylation at S15/20 and acetylation of K320 to induce cell cycle arrest genes (like *p21* and *GADD45*), and becomes a fully activated form through further modification of phosphorylation at S46 and acetylation of K373 to active apoptosis genes (like *BAX* and *PUMA*) [103,104,105]. The transactivation function of p73 is also influenced by post-translational regulatory pathways [106]. Compared to the canonical sequence of p53 response elements, overall the p53 noncanonical sequences are much less activated by direct overexpression experiments of each p53 family member under no stimulation condition. Maybe some of noncanonical sequences were easy to skip or some of them have binding affinity but no function such as half-site only. Some p53 mutants (p53 S46F and S121F) are super p53s and have much higher transactivation function than wild type [107,108], and p53 S46F also be used to assay for 4AT gap noncanonical sequence with much higher activity than wild type [23]. p73 has also been reported that contained super p73 mutants (S139F and S260N) [109], and these mutants may be good tools to do the functional assay to screen noncanonical sequences. How the signaling cascades modify each p53 member to activate noncanonical sequences is a big issue which will make clear the specific functions of each p53.

## Figures and Tables

**Figure 1 ijms-20-03681-f001:**
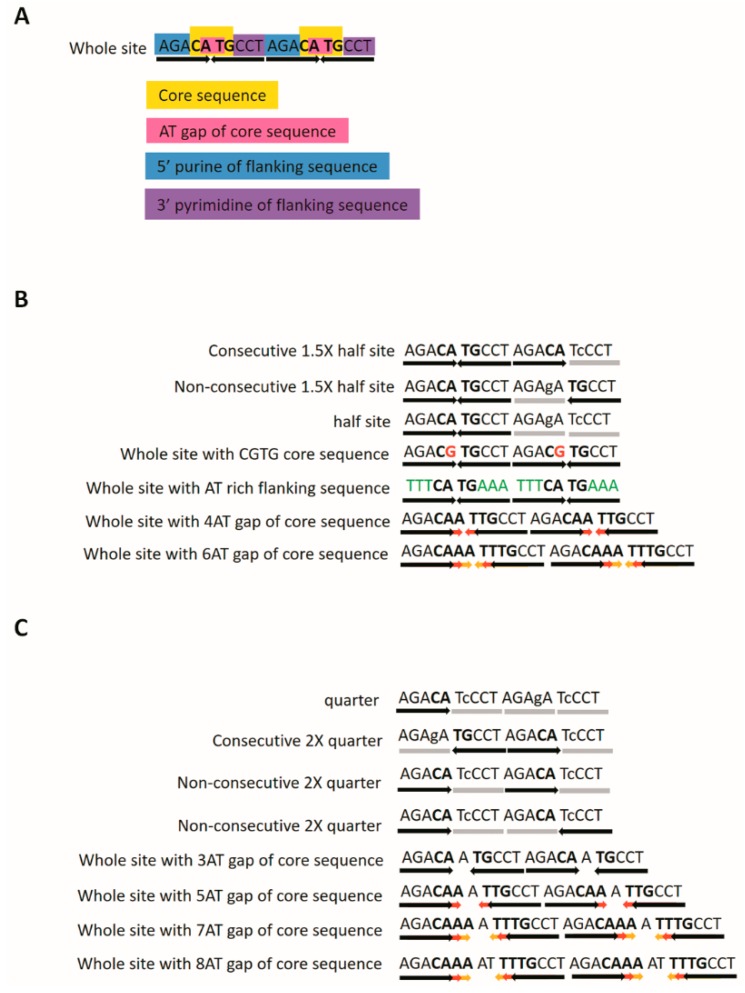
The noncanonical sequences of the p53 family regulation sites. (**A**) The canonical sequence is made up of repeats of 10mer as a whole-site which is composed of a 4-base core sequence and two flanking sequences as 5′ purine and 3′ pyrimidine (3 bases). The middle of the core sequence contains a 2-base AT gap. (**B**) Noncanonical sequences including different lengths of consensus sequence, variance of core or flanking sequences, and 4- or 6-base AT gaps. Red arrow: addition one A or T base in the canonical core sequence. Yellow arrow: addition second one A or T base in the canonical core sequence. Green word: substitution 5′ purine and 3′ pyrimidine for AT-rich of flanking sequences. grey straight line: non-functional quarter. (**C**) No effective sequences include different types of quarters and 3-, 5-, 7-, or 8-base AT gaps.

**Figure 2 ijms-20-03681-f002:**
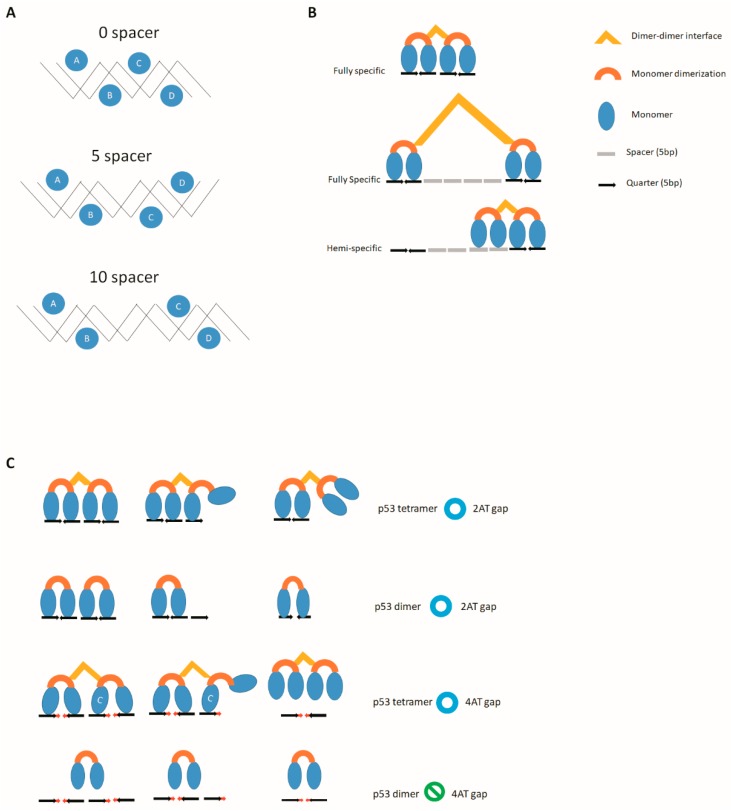
Binding pattern of noncanonical sequences (**A**) The first and third monomer A and C are in the same face of major groove when the responsive elements contain a 0 or 10 bp spacer. The first and third p53 monomer A and C are in the opposite face of the major groove when the response element contains a 5 bp spacer. (**B**) There are two tetramer binding patterns when the two half-sites are separated by 20 bp of spacer. In the fully specific pattern both dimers bind to the half-site. In the hemispecific pattern, one dimer binds to the half-site, and the other dimer binds to the spacer. (**C**) Both the tetrameric and dimeric p53 can bind to the whole-site, 1.5× half-site, and half-site with 2 AT gaps. Tetrameric p53 can bind to the whole-site and 1.5× half-site, but not half-site with 4 AT gaps, and dimeric p53 cannot bind to the whole-site, 1.5× half-site, or half-site. Therefore, the binding of the monomer C to the third quarter is essential to help the tetramer binding to the whole-site or the 1.5× half-site with 4 AT gaps.

**Table 1 ijms-20-03681-t001:** The motifs of p53 family responsive element sequences from prediction software (all raw data of frequency matrix are taken from the website indicated, and motifs are drawn by WebLogo [110]).

Software	Website		Reference
JASAPR	http://jaspar.genereg.net/		[87]
Members		Motif	
p53 model 1 (Motif ID: MA0106.1)		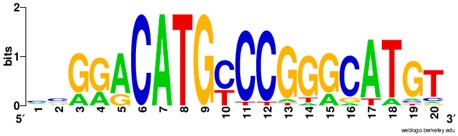	
p53 model 2 (Motif ID: MA0106.2)		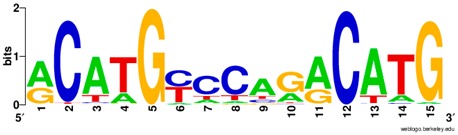	
p53 model 3 (Motif ID: MA0106.3)		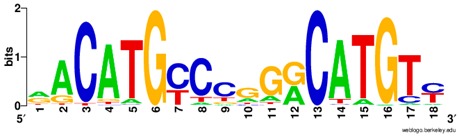	
p63 model 1 (Motif ID: MA0525.1)		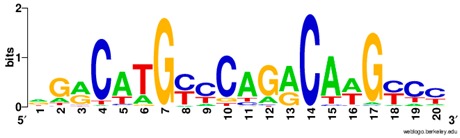	
p63 model 2 (Motif ID: MA0525.2)		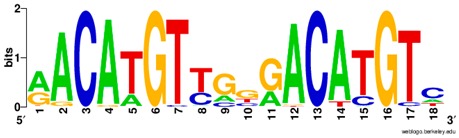	
p73 model (Motif ID: MA0861.1)		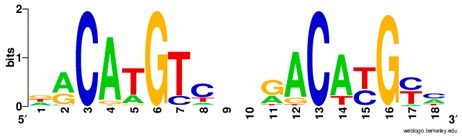	
Software	Website		Reference
CIS-BP	http://cisbp.ccbr.utoronto.ca/index.php		[111]
Members		Motif	
p53 model 1 (Motif ID: M09337_2.00)		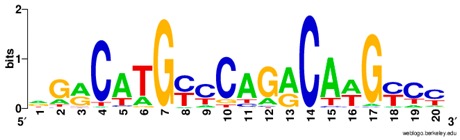	
p53 model 2 (Motif ID: M09621_2.00)		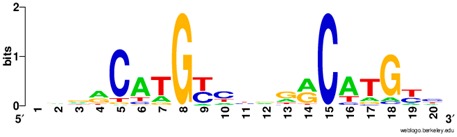	
p53 model 3 (Motif ID: M11197_2.00)		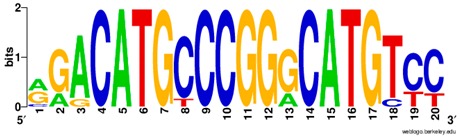	
p53 model 4 (Motif ID: M11198_2.00) * half-site only		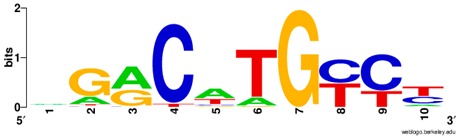	
p63 model 1 (Motif ID: M03437_2.00)		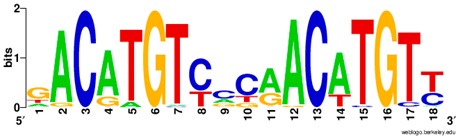	
p63 model 2 (Motif ID: M09335_2.00)		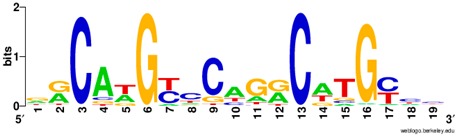	
p73 model 1 (Motif ID: M08035_2.00)		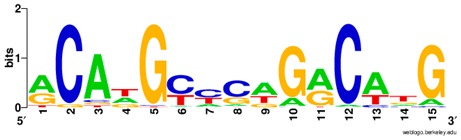	
p73 model 2 (Motif ID:M08036_2.00)		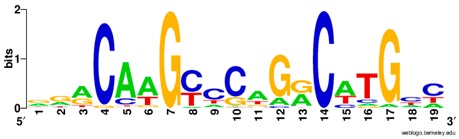	
p73 model 3 (Motif ID:M08037_2.00)		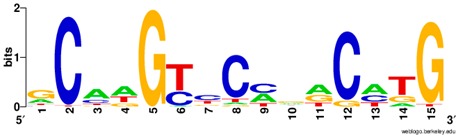	
p73 model 4 (Motif ID:M09336_2.00)		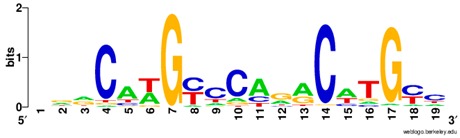	
Software	Website		Reference
HOCOMOCO	http://hocomoco11.autosome.ru/		[112]
Members		Motif	
p53 model 1 (Motif ID: P53_HUMAN.H11MO.0.A)		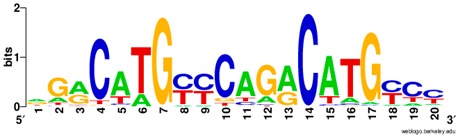	
p53 model 2 (Motif ID: P53_HUMAN.H11MO.1.A) * half-site only		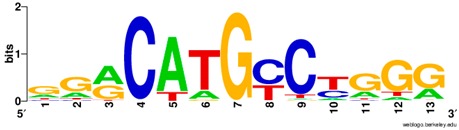	
p63 model 1 (Motif ID: P63_ HUMAN.H11MO.0.A)		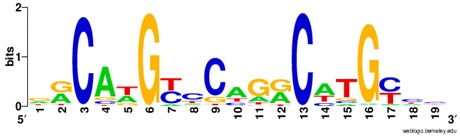	
p63 model 2 (Motif ID: P63_HUMAN.H11MO.1.A) * half-site only		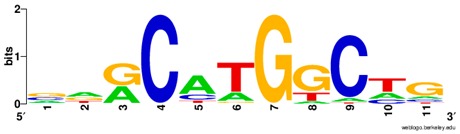	
p73 model 1 (Motif ID: P73_HUMAN.H11MO.0.A)		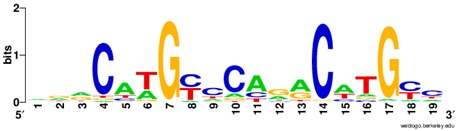	
p73 model 2 (Motif ID: P73_HUMAN.H11MO.1.A) * half-site only		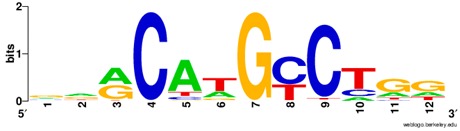	
Software	Website		Reference
PROMO	http://alggen.lsi.upc.es/cgi-bin/promo_v3/promo/promoinit.cgi?dirDB=TF_8.3		[113,114]
Members		Motif	
p53 model * half-site only		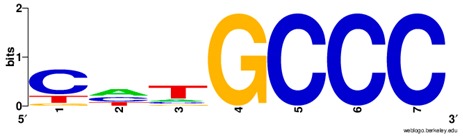	

**Table 2 ijms-20-03681-t002:** Specific target genes of each p53 family member and their responsive element sequences. (Functional definition is according to Fischer (2017): 1: Apoptosis; 2: Cell cycle arrest; 3: Autophagy; 4: Metabolism; 5: DNA repair; 6: Translational control; 7: Feedback mechanisms; 8: Others.)

Gene	Response Element Sequences	Noncanonical	p53	p63	p73	Function	Reference
*FUCA1*	+2609 GGG**CAAG**TTCATG**CAAG**TTC +2628		O	ND	X	3	[89,90]
*Dlx3*	−89 AAG**CA**AGA**CTTG**CAG −75	1.5× half-site	X	O	ND	8	[115]
*SMARCD3*	−226 GGG**CGTG**CAGATG**CAAG**CAC −207	CGTG core sequence	X	O	ND	8	[18]
*KRT14*	−140 AGA**CATG**ATG −131	half-site	X	O	ND	8	[16]
*ΔNp73*	−76 GGG**CAAG**CTGAGG**CCTG**CCC −57	CCTG core sequence	X	X	O	7	[116]
*NEU4*	−504 GGT**CCTG**GTCTCGT**CATG**CTT −484	CCTG core sequence	X	ND	O	3	[88]
*MLH3*	+ 56 GCG**CATG**CTC +65	half-site	X	ND	O	5	[117]
*G6PD*	+6085 GGT**CATG**AGCAAA**CATG**ACC +6104		X	X	O	3	[118,119,120]
*ITGB4*	+17 GCT**CCTG**CCCCGA**CAGG**TGC +36	CCTG and CAGG core sequence	X	O	O	8	[121,122]
*WNT4*	−141 GGG**CAGG**CTGCCGG**CAGG**CAC −121	two CAGG core sequence	X	O	O	8	[123]
*MDR1*	−210 CTA**CTTG**CCCTTT**CTAG**AGA −191	Second half-site with 5′ PyPyPy and 3′ PuPuPu flanking sequence	X	O	O	4	[72]
*apoD*	−419 ATA**CCAG**ATGTTTGAAAA**CATG** TTGCAACACGTC**CTGC**TG −380	CCAG and CCTG core sequence	X	O	O	4	[124]
*FLOT2*	−4029 GGA**CTTG**GCCAGT**CTGG**CCT −4010	CTGG core sequence	X	O	O	8	[125]
*IAPP*	−1756 CAA**CATG**AGGCTG**CATG**TCA −1763 (site 1) mouse genome +706 GAA**CATG**TTTTAGGACATA**CAGG** GGC +732 (site 2) mouse genome	First half-site with a 3′ PuPuPu flanking sequence (site 1) CAGG core sequence (site 2)	X	O	O	3	[126]
*JAG1*	*+5597* AGG**CT**TCTTGTTCAGG**CTTG**CTCT GTGTGAA**CCAG**ACCGTTGTG**CTTG**GCT *+5647*	A 1.5× half-site plus a whole-site with a CCAG core sequence	X	O	O	8	[10,127]
*PEDF*	−1366 AAA**CTTG**TTTTAAAA**CAAG**CTTG TGTAACC**CATG**ACC −1330	AT-rich of flanking sequence of first and second core sequence	X	O	O	1	[128]

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
