# Peer review of "Roles of p53 Family Structure and Function in Non-Canonical Response Element Binding and Activation"

_ijms, 2019, doi:10.3390/ijms20153681_

Reviewer 1 Report

The review by Cai et al. describes the structure and regulation of non-canonical p53 binding sites. In general, the manuscript gives a broad overview of possible variations of non-canonical binding sites with special emphasis on the regulation of transactivation by p53 family members. The paper gives an extensive overview with a significant amount of detail that will will certainly be of interest to experts in the field. To make it more accessible for a broader readership, a few extra points should be addressed:

 1.     Before describing the details of p53 family members binding to non-canonical p53 binding sequences, the authors should briefly explain what the differences are compared to canonical response elements (also from a functional point of view), where can such sites be found and how many genes may contain consensus or non-canonical sites.

2.     In a kind of introduction, it should also be stated when (under which conditions) the different sites canonical vs. non-canonical) become activated to what extent.

3.     In general, putting the information in a more functional context would definitely help.

4.     Finally, the authors need to improve their English. Many phrases cannot be understood.

 Author Response

The review by Cai et al. describes the structure and regulation of non-canonical p53 binding sites. In general, the manuscript gives a broad overview of possible variations of non-canonical binding sites with special emphasis on the regulation of transactivation by p53 family members. The paper gives an extensive overview with a significant amount of detail that will will certainly be of interest to experts in the field. To make it more accessible for a broader readership, a few extra points should be addressed:

 1.     Before describing the details of p53 family members binding to non-canonical p53 binding sequences, the authors should briefly explain what the differences are compared to canonical response elements (also from a functional point of view), where can such sites be found and how many genes may contain consensus or non-canonical sites.

We have provided more information and a website on pages 1 and 2.

2.     In a kind of introduction, it should also be stated when (under which conditions) the different sites canonical vs. non-canonical) become activated to what extent.

 The non-canonical sequences were assayed by binding or over-expression system only, the signaling cascades for modification of each p53 member to active non-canonical sequences is still a mystery. We have added some discussion in the end of the article.

 3.     In general, putting the information in a more functional context would definitely help.

We added the function of genes in Table 2.

4.     Finally, the authors need to improve their English. Many phrases cannot be understood.

We have had the English in the revised manuscript checked by a Board of Editors in the Life Sciences (BELS)-certified editor whose mother tongue is English.Reviewer 2 Report

This is a well written review covering the topic stated in the title.

Author Response

This is a well written review covering the topic stated in the title.

Thank you very much.  The English in the revised manuscript was thoroughly checked by a native speaker of English.